# ROUTERBENCH: A Benchmark for Multi-LLM Routing System

Qitian Jason Hu [1]   Jacob Bieker [1]   Xiuyu Li [2]   Nan Jiang [3]   Benjamin Keigwin [1]   Gaurav Ranganath [1]
Kurt Keutzer [2]   Shriyash Kaustubh Upadhyay [1]

## Abstract

As the range of applications for Large Language Models (LLMs) continues to grow, the demand for effective serving solutions becomes increasingly critical. Despite the versatility of LLMs, no single model can optimally address all tasks and applications, particularly when balancing performance with cost. This limitation has led to the development of LLM routing systems, which combine the strengths of various models to overcome the constraints of individual LLMs. Yet, the absence of a standardized benchmark for evaluating the performance of LLM routers hinders progress in this area. To bridge this gap, we present ROUTERBENCH, a novel evaluation framework designed to systematically assess the efficacy of LLM routing systems, along with a comprehensive dataset comprising over 405k inference outcomes from representative LLMs to support the development of routing strategies. We further propose a theoretical framework for LLM routing, and deliver a comparative analysis of various routing approaches through ROUTERBENCH, highlighting their potentials and limitations within our evaluation framework. This work not only formalizes and advances the development of LLM routing systems but also sets a standard for their assessment, paving the way for more accessible and economically viable LLM deployments. The code and data are available at https://github.com/withmartian/routerbench.

## 1. Introduction

Large Language Models (LLMs) have exhibited remarkable capabilities in addressing a wide range of tasks across academic and industrial scenarios (Bubeck et al., 2023). This has motivated both researchers and practitioners to intro-duce new LLMs, designed for both generic and specialized use cases, on a near-daily basis [1]. However, the proliferation of LLMs presents a challenge for LLM application builders to identify the most suitable model for their applications. While some proprietary models, such as GPT-4, are distinguished by their superior performance, they often incur high economic costs due to the high API prices.

Many prior works focus on improving the capabilities of individual LLMs while maintaining low costs. Techniques such as prompting (Wei et al., 2022), quantization (Lin et al., 2023; Kim et al., 2023), and system optimization (Kwon et al., 2023) may reduce a single model's serving cost, yet with new models emerging daily, these approaches may not remain feasible or scalable in long term. Moreover, the diversity of choices of LLMs available at various price and performance tiers can be daunting for users attempting to select and optimize an appropriate model[2].

An alternative solution aims to select to optimal LLM for each input through "routing." (Chen et al., 2023; Shnitzer et al., 2023; Šakota et al., 2023). Routing offers several advantages over single-LLM optimization. First, it is a lightweight process, which treats each LLM as an input-output black box, avoiding the need to delve into intricate infrastructure details, thus making it flexible and broadly applicable. Second, routing systems benefit from the diversity of LLMs, while single-LLM methods may struggle to keep pace with the expanding LLM landscape. Lastly, while single-LLM strategies often face a compromise between performance and other factors such as per-token costs, routing systems adeptly balance a spectrum of user demands.

The rise in routing-related research has improved cost efficiency, enhanced performance, and broadened accessibility (Chen et al., 2023; Lee et al., 2023; Lu et al., 2023). Despite these advances, a comprehensive benchmark for evaluating routing techniques remains absent. We introduce ROUTERBENCH, the first comprehensive benchmark designed specifically for assessing router mechanisms in terms of inference dollar cost and performance.

---

[1]Martian [2]UC Berkeley [3]UC San Diego. Correspondence to: Qitian Jason Hu <jason@withmartian.com>.

Preprint.

[1]As of January 16th, 2024, there are 469,848 models listed on huggingface.com

[2]As of January 29th, 2024, there are 22,177 language models with 7 billion parameters listed on huggingface.com

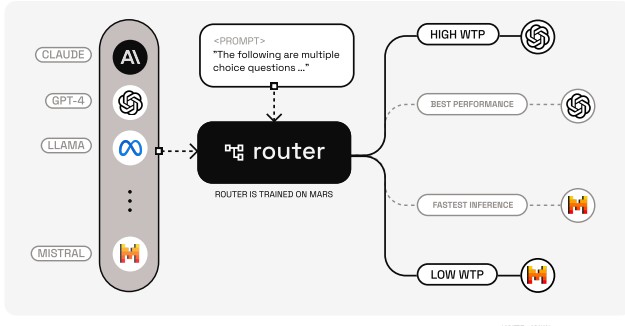

*Figure 1.* **Left**: The ROUTERBENCH Construction Process integrates eight datasets with eleven distinct models to develop ROUTERBENCH. Detailed format can be found in Appendix A.3. **Right**: The Model Routing Process shows the method of routing prompts through a router to various LLMs based on specific requests, demonstrating the dynamic allocation of resources.

ROUTERBENCH encompasses a diverse array of tasks and domains, with pre-generated LLM response and quality metrics, on which different routing mechanisms can be efficiently tested without inference. Our experiments revealed that while some previous routing mechanisms have difficulty generalizing to complex tasks and up-to-date models, there are several promising fields on which even simple routing demonstrated outstanding performance.

In conclusion, we present the following key contributions:

1. We have developed a comprehensive benchmark for LLM routing covering major tasks for LLMs, which includes a wide range of both open-source and proprietary models. ROUTERBENCH enables efficient training and testing of model routers without inference, and can be flexibly extended to cover new tasks and models.

2. We introduce a theoretical framework designed to assess the efficacy of routers across several metrics, with a particular emphasis on inference cost (expressed in dollars) and performance. This framework includes mathematical formulations that enable the seamless integration and comparative analysis of various routers and LLMs.

3. We evaluate the efficiency of routing strategies across a broad range of tasks. Our results provide insights into the performance of various routers in different contexts and demonstrate that the monetary costs of LLM services can routinely vary by factors of 2-5× for comparable levels of performance. This underscores the significance and utility of our benchmark, highlighting promising areas for future enhancements.

## 2. Related Work

Various strategies have been proposed to optimize the cost and performance of current LLMs. We provide an overview of them with a focus on routing-related approaches.

**Single LLM Enhancement** Fine-tuning is used to improve models for specific tasks, which requires additional training and domain-specific data (Rafailov et al., 2023). Prompting mechanisms like Chain-of-Thought (CoT) (Wei et al., 2022; Zhou et al., 2023; Wang et al., 2022) and Tree of Thoughts (ToT) (Yao et al., 2023) could bolster LLM performance without additional fine-tuning. Mixture-of-Experts (MoE) (Eigen et al., 2014; Shazeer et al., 2017; Fedus et al., 2022; Du et al., 2022; Shen et al., 2023; Si et al., 2023) is another line of work that explores routing within the model to enhance performance efficiently, which contains specialized "experts" and routes the input to the best expert. Nevertheless, these single-LLM enhancements are usually model and scenario specific, and could not benefit from the explosion in the number of LLMs.

**LLM Synthesis** Beyond single LLM approaches, LLM synthesis utilizes the ensemble of multiple LLMs, integrating their outputs into an enhanced final result (Jiang et al., 2023b). Another approach has shown that a strategic combination of smaller models can match or even outperform larger models (Lu et al., 2024). However, these methods require at least two steps: text generation and synthesis, which increases costs and latency, creating challenges to applying this approach in production.

**Routing** Unlike LLM Synthesis, routing can select the suitable model for specific input without performing inference on every candidate model. Routing can be classified into two categories, non-predictive routing and predictive routing. Non-predictive routing strategies retrieve outputs from LLMs and directly pick one without a model-assisted syn-

thesis step. FrugalGPT (Chen et al., 2023) presents an inaugural application of this type of strategy, which employs a generation judger that assesses the quality of responses from various LLMs to a given query, invoking LLMs sequentially until an answer meets a predefined quality threshold. Several studies (Madaan et al., 2023; Yue et al., 2023; Lee et al., 2023) also have explored systems that integrate small language models with LLMs. Another methodology involves a layered inference framework, re-routing more complex queries to an advanced model for improved results (Wang et al., 2023). Predictive routing selects the optimal LLM without requiring to evaluate the output. One line of research has implemented routers utilizing supervised learning algorithms (Shnitzer et al., 2023), while some others use reward model-based techniques (Hari & Thomson, 2023; Lu et al., 2023). Furthermore, meta-model, trained on inputs along with a model-specific token to predict the performance score, represents another approach to determining the most appropriate LLM for use (Šakota et al., 2023). In short, predictive routers could bring substantial cost and performance improvement without sacrificing latency, with a number of early works dedicated to this field.

While many routers currently exist, a systematic benchmark for their evaluation has been lacking. Our work aims to address this issue and introduce a benchmark for router evaluation.

## 3. Math Formulation for Router Evaluation

The primary challenge in assessing the performance of routing systems lies in balancing two conflicting objectives: maximizing efficiency and minimizing cost. To effectively compare routers, we have developed a framework that captures the multi-faceted nature with one metric.

### 3.1. Setup and Basic Operations

Consider a set of models $L = \{LLM_1, \ldots, LLM_m\}$ a dataset $D$ consisting of examples $x_i \in \{x_1, ..., x_{|D|}\}$. For each model $LLM_j$, we evaluate its performance by generating an output $o_i^j = LLM_j(x_i)$ for each example $x_i$. Each output $o_i^j$ has two associated quantities: the cost $c(o_i^j)$ of generating that output and the quality or performance $q(o_i^j)$ of the output itself. Through this process, we establish an expected cost $c_m$ and an expected quality $q_m$ for each model $LLM_m$ across the dataset $D$.

$$c_m = E[c(LLM_m(x))|x \in D]$$

$$q_m = E[q(LLM_m(x))|x \in D]$$

A *router $R$*, define as a function, takes in a prompt $x$ and a set of parameters $\theta$, subsequently selecting the most suitable model $LLM_i$ from a set $L$ to complete the prompt, i.e.

$$R_\theta(x) \mapsto LLM_i \in L.$$

The parameters $\theta$ typically include the maximum price the user is willing to pay, the desired latency, or a number of layers of neural networks for the router model, etc. More details of router parameters will be elaborated and discussed in Section 5.1.

The expected cost of a router $R_{\theta_1}$ across dataset $D$ is defined as

$$c_{R_{\theta_1}}(D) = E[c(R_{\theta_1}(x))|x \in D]$$

and the expected performance of a router $R_{\theta_1}$ can be defined similarly.

By experimenting with various router parameters $\theta_1, ..., \theta_k$, we obtain a series of data points $(c_{R_{\theta_1}}, q_{R_{\theta_1}}), ..., (c_{R_{\theta_k}}, q_{R_{\theta_k}})$ which can be graphically represented in the cost-quality $(c - q)$ plane alongside the results of LLMs for comparative analysis.

**Linear Interpolation** The initial operation we introduce within this framework is *linear interpolation*, which enables the computation of a weighted average between any two points on the cost-quality $(c - q)$ plane.

As illustrated by an example in the left of Figure 2, consider two routers, $R_{\theta_1}$ and $R_{\theta_2}$, we can formulate a third router, $R_{int}(R_{\theta_1}, R_{\theta_2})$, based on the following principle: given a prompt $x$ select $t \in [0, 1]$ such that:

$$R_{int}(R_{\theta_1}, R_{\theta_2}, t)(x) = \begin{cases} R_{\theta_1}(x), & \text{w.p. } t \\ R_{\theta_2}(x), & \text{w.p. } 1 - t \end{cases}$$

Through the principle of linearity of expectation, we can deduce the expected cost of $R_{int}(R_{\theta_1}, R_{\theta_2}, t)(x)$ in terms of $LLM_1$ and $LLM_2$:

$$E[c_{R_{int}(x)}|x \in D] = t \cdot c_{R_{\theta_1}} + (1 - t) \cdot c_{R_{\theta_2}}$$

and the expected performance of $R_{int}(R_{\theta_1}, R_{\theta_2}, t)(x)$ can be defined similarly.

Notably, for two data points $(c_1, q_1)$ and $(c_2, q_2)$ corresponding to $R_{\theta_1}$ and $R_{\theta_2}$ respectively, $R_{int}(t)$ can precisely interpolate any point along the line segment connecting $(c_1, q_1)$ and $(c_2, q_2)$.

**Extrapolation** To ensure all routers can enrich our mathematical framework, we also introduce the *extrapolation* operation, which enables all routers to extend to the cost domain $[0, \infty]$. For a given router $R_\theta$, we can trivially add more cost to the system without adding performance (for example repeat LLM generation $k$ times and only take the last generation as final output) and thus extend the cost to $\infty$. To extend the router to a smaller cost domain, we simply interpolate the null router (zero cost, zero performance) and $R_{\theta_1}$. Thus we are able to achieve any cost level between $[0, \infty]$ to when comparing routers with different domains.

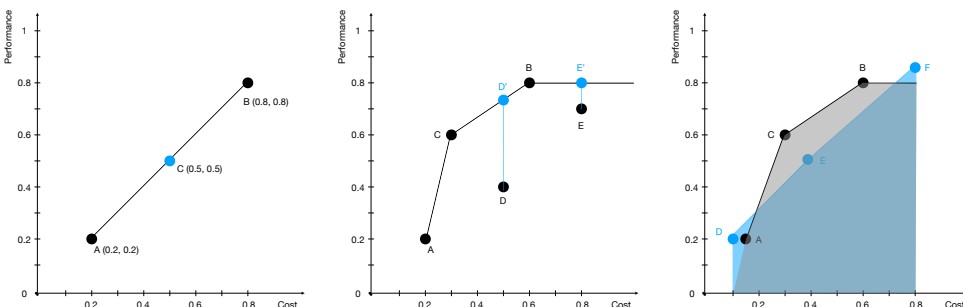

*Figure 2.* **Left**: linear interpolation is the process of achieving the cost-performance trade-off between any concrete routers. Point A and B are routers with different input parameters. To achieve the average of A and B, we build router C which routes to A or B with 50% probability each, and it performs the average of A and B in expectation. **Middle**: Consider points A to E, we can construct the non-decreasing convex hull consisting of points A, B, and C. D and E as they can be replaced by a strictly superior affine combination of A, B, and C. **Right**: ABC and DEF are two routing systems (already convexified with ABC extrapolated to (0.1,0) for a fair comparison). To compare, we interpolate A and B to $c_{min} = 0.1$ and $c_{max} = 0.8$, respectively, and then calculate the area under the curve normalized by $c_{max} - c_{min}$ to derive AIQ.

It is essential to note that the routers discussed are functionally analogous to LLMs within this context, as both can be represented as coordinates in the cost-quality $(c - q)$ plane.

### 3.2. Non-Decreasing Convex Hull

When working with multiple routers, it's feasible to construct any affine combination of points through linear interpolation among them. Specifically, for a set $S$ of points in the cost-quality $(c - q)$ plane, these affine combinations can target any point $(c, q)$ in $\mathbb{R}^2$ lying within the convex hull formed by $S$. We identify $S_{ch} \subseteq S$ as the subset of points that delineate the vertices of this convex hull.

Furthermore, it's possible to configure a non-decreasing convex hull from $S_{ch}$, ensuring that for any two points $(c_1, q_1)$ and $(c_2, q_2)$ where $c_2 \geq c_1$, it follows that $q_2 \geq q_1$. Intuitively, if the extra cost of $c_2 - c_1$ does not bring any performance improvement, it is advisable to simply extrapolate $(c_1, q_1)$ to the domain of $c_2$, and $(c_2, q_2)$ could be $(c_2, q_1)$. An example is shown in the middle of Figure 2.

For a given routing system $R_1$, constituted by LLMs and routers plotted in the $c - q$ plane for dataset $D$, we can conceptualize a new routing system $\widetilde{R_1}$. This involves constructing routers $R_{\theta_1}, ..., R_{\theta_k}$, yielding points $(c_1, q_1), ..., (c_k, q_k)$. By establishing a non-decreasing convex hull $S_{ndch}$ from these points and for any cost $c$ within the range $[c_{min}, c_{max}]$, optimal performance is attainable by interpolating between the two closest cost points. This process effectively creates a new routing system that spans the complete domain $[c_{min}, c_{max}]$.

Given the framework established, we define the **Zero router** ($R_{zero}$) as a router that selects LLMs from $\{LLM_1, \ldots, LLM_m\}$ based on their collective non-decreasing convex hull. For a specified cost $c$, $R_{zero}$ provides a probabilistic mix of LLMs that maximizes expected output quality with a simple, mathematics-driven routing strategy. $R_{zero}$ serves as a basic benchmark for assessing the efficacy of other routing systems; a router is deemed significant only if it demonstrates superior performance compared to $R_{zero}$.

### 3.3. Comparing Different Routing Systems

Given the agnostic nature of our comparison framework towards the router's structure, routing systems can produce an assorted set of points on the $c - q$ plane that may be non-deterministic and non-parametric, complicating direct comparisons. Leveraging the methodologies delineated previously, we have the capacity to condense these disparate points into a streamlined function—specifically, a non-decreasing convex hull—and subsequently distill this representation into a singular metric that encapsulates the system's characteristics.

Routing systems often generate multiple points on the cost-quality $(c - q)$ plane, making it difficult to compare the underlying systems. However, our framework allows us to transform these non-parametric points into a simpler function, specifically a non-decreasing convex hull, which can be characterized by a single numerical value.

Let's consider two different routing systems (for example KNN and MLP-based routers), $R_\theta$ where $\theta \in \Theta$, and $R_\lambda$ where $\lambda \in \Lambda$. To compare their effectiveness, we parametrize them by sampling from $\Theta, \Lambda$ to generate a set of points: $R_{\theta_1}, \ldots, R_{\theta_k}$, and $R_{\lambda_1}, \ldots, R_{\lambda_k}$. Then, we construct a non-decreasing convex hull for both groups, $\widetilde{R_\theta}$ and

$\widetilde{R_\lambda}$, defined on a shared domain $[c_{min}, c_{max}]$.

We define $AIQ$ (Average Improvement in Quality) for one of the routing systems as follows:

$$AIQ(R_\theta) = \frac{1}{c_{max} - c_{min}} \int_{c_{min}}^{c_{max}} \widetilde{R_\theta} \, \mathrm{d}c$$

With the equation above, we can calculate AIQs for any group of routing systems to get a clear understanding of their relative performance, which is demonstrated in the right of Figure 2. Rather than performing complex graphic analysis, $AIQ$ allows users to measure router performance in a straightforward way.

## 4. Benchmark Construction - ROUTERBENCH

To systematically assess router performance, we have developed a dataset, ROUTERBENCH. This comprehensive dataset consists of a broad spectrum of tasks, including commonsense reasoning, knowledge-based language understanding, conversation, math, coding and retrieval-augmented generation (RAG). ROUTERBENCH is constructed by leveraging existing datasets that are widely recognized and utilized in the evaluation of leading LLMs, such as GPT-4, Gemini (Team et al., 2023), and Claude (Anthropic, 2023). This approach ensures that ROUTERBENCH is representative of the diverse challenges and requirements pertinent to mainstream LLM performance evaluation.

### 4.1. Principles in benchmark construction

The construction of ROUTERBENCH is guided by the following principles:

- Extensive Coverage: Our selection process identified a diverse array of fields where LLMs are widely utilized, aiming for wide-ranging applicability.

- Practical Relevance: The benchmarks chosen are of considerable significance to the industry's current applications of LLM systems, presenting a balanced challenge to the state-of-the-art LLMs, that is not too difficult nor too simplistic.

- Extensibility: ROUTERBENCH is designed for seamless integration of additional metrics, such as latencies and throughputs, ensuring adaptability to the evolving landscape of LLM.

### 4.2. Benchmark Dataset

For the initial release, we have curated a selection of 8 representative datasets from multiple different tasks. Detailed descriptions are in Appendix A.2.

- **Commonsense Reasoning**: Hellaswag (Zellers et al., 2019), Winogrande (Sakaguchi et al., 2021), and ARC Challenge (Clark et al., 2018)

- **Knowledge-based Language Understanding**: MMLU (Hendrycks et al., 2021)

- **Conversation**: MT-Bench (Zheng et al., 2023b)

- **Math**: GSM8K (Cobbe et al., 2021)

- **Coding**: MBPP (Austin et al., 2021)

**RAG Dataset**: To evaluate routers in a more practical setting, we collected 800 user queries from one of Martian's clients, an LLM-assisted search company, and constructed an RAG dataset based on these queries. These queries cover topics including sports, history, media & art, and politics, and all have ground truth answers. We then manually collected the ground truths, which were used to evaluate answers from selected groups of LLM and LLM-assisted search engines. This initiative is designed to assess the routers' performance in a complex "compound system" setting (Zaharia et al., 2024) – determining whether routers can adeptly navigate when retrieval abilities are also in play. For instance, when dealing with news published after the GPT-4 knowledge cutoff, routers are expected to more frequently opt for models that can access and search the latest internet-based information (e.g. sonar-medium-online).

### 4.3. Dataset Construction Process

For the compilation of our benchmark dataset, we perform inference with 14 different LLMs, with 3 of them specific to the RAG dataset[3], including both open-source and proprietary models. This process was applied to each of the eight datasets and the RAG dataset enumerated in Section 4.2, which is also illustrated in Figure 1. The selected LLMs are as follows and more details are in Appendix A.1:

**Open Source Model:** Llama-70B-chat (Touvron et al., 2023), Mixtral-8x7B-chat (Madaan et al., 2023), Yi-34B-chat (AI et al., 2024), Code Llama-34B (Rozière et al., 2023), Mistral-7B-chat (Jiang et al., 2023a), WizardLM-13B (Xu et al., 2024)

**Proprietary Model:** GPT-4, GPT-3.5-turbo (OpenAI, 2023), Claude-instant-v1, Claude-v1, Claude-v2 (Anthropic, 2023), You.com API, sonar-small-online, sonar-medium-online.

In total, there are 405,467 samples in ROUTERBENCH, covering 11 models, 8 datasets, and 64 tasks.

---

[3]sonar-small-online and sonar-medium-online from Perplexity AI, You.com API

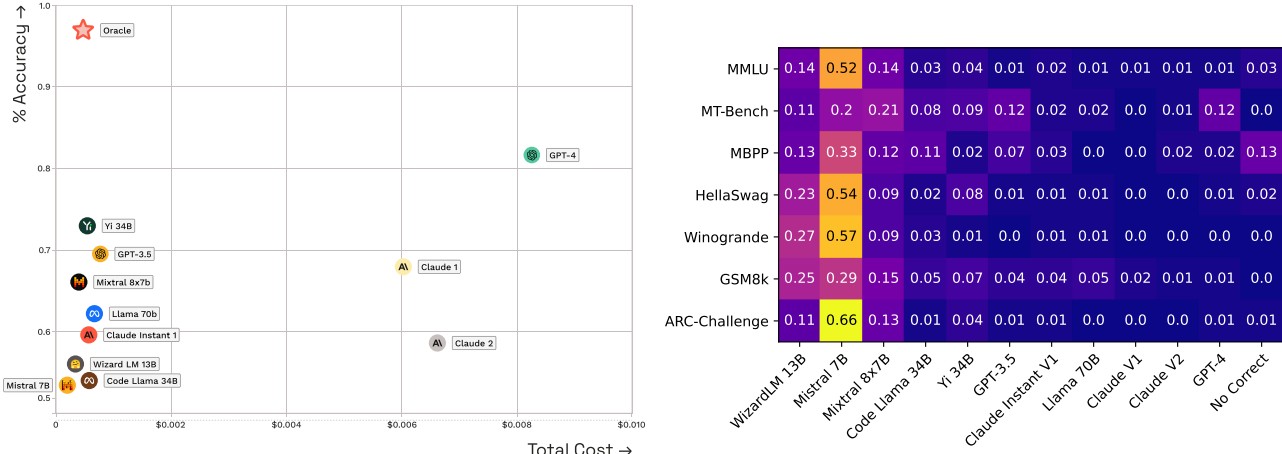

*Figure 3.* **Left**: Accuracy vs Total Cost of all the 11 LLMs on ROUTERBENCH. **Right**: The *Oracle* LLMs selection frequency across the 7 subsets in ROUTERBENCH.

### 4.4. A Pilot Study: The Oracle Router

We assessed the performance of various models across the eight datasets, with more details in ( A.4 and 7) while aggregate results are illustrated in Figure 3. The *Oracle* represents the best possible router: the one that always routes to the best-performing LLM (if there are multiple of them, then route to the cheapest one).

**Result:** We note that the *Oracle* router achieves near-optimal performance at a low cost, highlighting the potential for efficient routing among LLMs. Although proprietary models like GPT-4 offer superior performance, their higher cost than open-source alternatives is a significant drawback. Factors such as overalignment could also hurt the generation quality of proprietary models such as Claude 2 (refer to Appendix C). The heatmap in Figure 3 illustrates that, despite WizardLM-13B and Mistral-7B achieving only about 50% accuracy across tasks, their affordability leads to frequent selection by the *Oracle*, prioritizing them when they provide correct responses. Moreover, the surprising observation that GPT-4 is seldom chosen suggests the existence of less expensive LLMs that can deliver high-quality answers for most queries. This underscores the substantial opportunity for enhancing LLM systems through cost-effective routing without sacrificing quality.

## 5. Experiments

### 5.1. Predictive Router

We propose a novel set of predictive routers that do not require the pre-generation of LLM outputs. Specifically, we introduce a router $R : x_i \rightarrow$ LLM, constructed as follows: for an input $x_i$, the performance score for $LLM_j$ is

calculated via:

$$\text{performance score}_{ij} = \lambda \cdot P_{ij} - \text{cost}_j$$

$P$ denotes the predicted performance of $LLM_j$ on sample $x_i$, with $\lambda$ representing the **willingness to pay (WTP)** parameter that delineates the cost-performance trade-off. A higher $\lambda$ indicates a preference for superior performance at a higher cost. We approximate the total cost using the cost per token metric. The routing decision for the predictive router is thus formulated as selecting the $LLM$ that optimizes the performance score.

To estimate $P$ for each input across models, we implemented two supervised regression approaches: **k-nearest neighbors (KNN)** and **multi-layer perceptron (MLP)** inspired by (Shnitzer et al., 2023). We allocated a fraction of the dataset for training a performance predictor for each task, assessing its efficacy on the remainder.

Specifically, the **KNN router** estimates performance score$_{ij}$ by identifying the $k$ nearest samples in the training set $D_{train}$ and opting for $LLM_i$, demonstrating optimal performance within this subset.

$$P_{\text{KNN}}(x_i) = \frac{1}{k} \sum_{x_j \in \text{NN}_k(x_i, D_{train})} q(o_j^i)$$

Where $NN_k(x_i, D_{train})$ signifies the subset of $k$ nearest neighbors to the sample $x_i$ within the training dataset $D_{train}$.

Similarly, for **the MLP router**, we have trained a set of MLP models to predict the performance

$$P_{\text{MLP}}(x_i) = f(W_n \cdot \sigma(... \cdot \sigma(W_1 \cdot x_i + b_1)... + b_n)$$

Those series of KNN and MLP routers are trained with varying hyperparameters, and we present the experimental

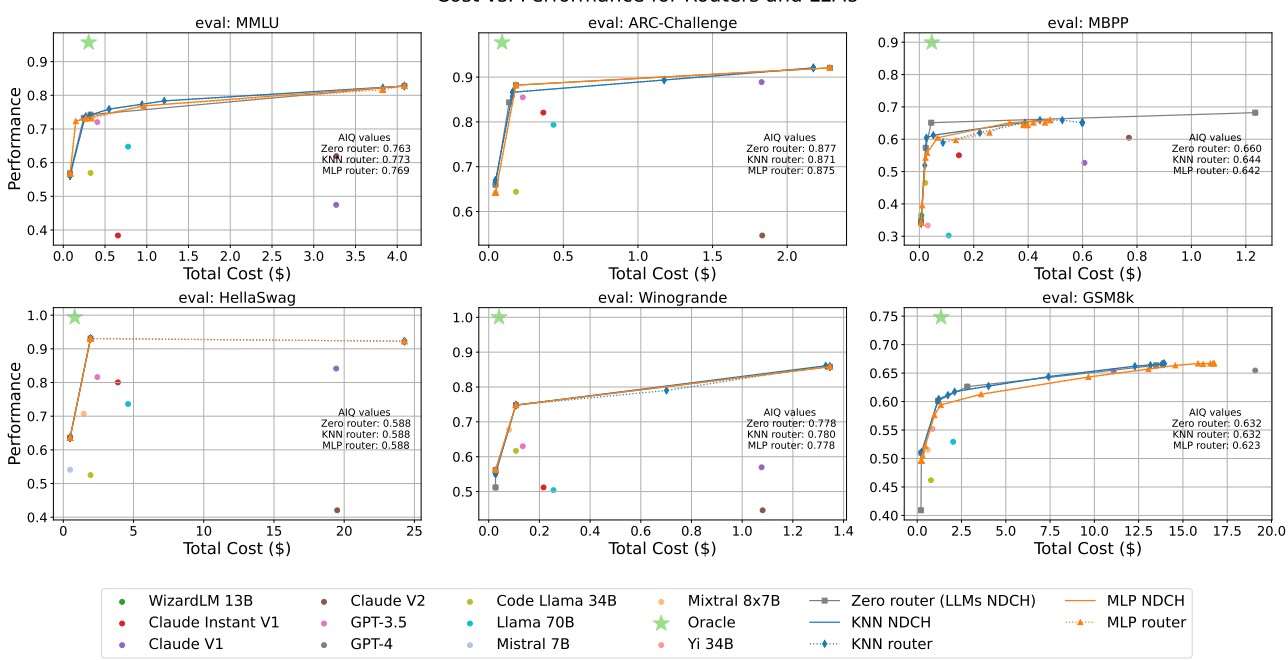

*Figure 4.* Total Cost vs. Performance for eleven models and KNN, MLP, and Zero routers on ROUTERBENCHexcept for MT-Bench. For KNN and MLP, we tested different hyper-parameters, and the optimal results are displayed above. The AIQ values are calculated for all 3 routers. NDCH stands for non-decreasing convex hull, represented by the solid lines. Dotted lines connect points with increasing willingness to pay.

results derived from the optimal hyperparameter configurations.

### 5.2. Non-Predictive Routers

This category of routers generates answers from a sequence of Large Language Models (LLMs), evaluates these answers, and bases routing decisions on the evaluation outcomes. Drawing inspiration from (Chen et al., 2023; Wang et al., 2023), we introduce a *cascading router* comprising of a total cost parameter $T$, and a sequence of $m$ LLMs, denoted as $LLM_j$ : text $\rightarrow$ text, ranked from the least to the most expensive in terms of computational cost and expected accuracy. A key component of its operation is a scoring function $g$ : text $\rightarrow [0, 1]$ paired with a threshold $t$ (the "judge"). Upon receiving a request, it is initially processed by $LLM_1$. If g(o1)¿t, the output o1 is selected, and the process terminates; otherwise, if the cumulative cost is still less than the total cost T, the router proceeds to the next LLM in the sequence and returns the current output if not.

Although developing an effective scoring function $g$ for a specific task in a production setting presents challenges, within the context of this paper, the router possesses perfect knowledge of the final score, enabling it to consistently select the most cost-effective model that yields a satisfactory response (akin to an oracle). To simulate real-world performance more accurately, we introduce an error parameter

$\epsilon \in [0, 1]$. The adjusted scoring function $g_\epsilon(o)$ is defined as:

$$g_\epsilon(o) = \begin{cases} 1 - g(o) & \text{with probability } \epsilon \\ g(o) & \text{with probability } 1 - \epsilon \end{cases}$$

A variant of the non-predictive router is overgenerate-and-rerank, which generates all potential outcomes from the LLM, assesses each, and outputs the optimal one as determined by a designated reward function. Although its practical application is limited due to significant costs, we will present its results for demonstration.

### 5.3. Main Results

**Predictive Router** With the KNN and MLP router design, we present the performances of predictive routers across all tasks (other than MT-Bench). The dataset for each task is randomly partitioned into two splits, where the routers are trained on 70% and evaluated on the rest 30%. We exclude MT-Bench in this set of experiments due to its limited size in performing such a train-test partition. As shown in Figure 4, both KNN routers and MLP routers achieve the level of performance to the best individual LLMs with lower or similar costs, demonstrating the effectiveness of the proposed routing solutions, despite their simplicity. However, none of the routing algorithms significantly outperform the baseline Zero router (The routers exhibit higher AIQ than the Zero router for MMLU and Winogrande, achieved com-

*Figure 5.* Total Cost vs Performance for eleven models and cascading routers on MMLU, MBPP, and GSM8K. Different error rates are tested, and the AIQ value is computed for Zero Router and zero error rate cascading router. The solid lines represent the non-decreasing convex hull and the dotted line represents points with increasing the maximum cost parameter.

parable AIQ for Hellaswag and GSM8K, and underperform on ARC-Challenge and MBPP), the *oracle* router consistently exceeds all other routers and LLMs in performance, underscoring the room for further advancements in routing algorithms design.

**Cascading Router** We present results for cascading routers on MMLU, MBPP, and GSM8K in Figure 5. The results indicate that with each error rate, as the total cost $T$ increases, the cascading router's performance improves due to the availability of a larger budget for selecting more appropriate models. For lower error rates, the cascading router performs better than the Zero router, as evidenced by the higher AIQ value. The router with a zero error rate judge quickly approximates the performance of the *Oracle* at the same cost and achieves comparable results as the cost further increases. Figure 5 illustrates the cascading routers' effectiveness, showing they surpass both individual LLMs and the Zero router by a significant margin when the router's judge has an error rate of up to $0.1$. This indicates the routing technique's potential when paired with an effective judge.

However, as the judge's error rates increase, the performance of the cascading router may deteriorate rapidly, particularly when the error rate exceeds $0.2$. Achieving a sufficiently low error rate for certain real-world tasks to benefit from cascading routers might be challenging. Additionally, the sequence in which LLMs are chosen plays a crucial role in performance and offers room for optimization (Chen et al., 2023). Our findings present a simulated upper limit for this method, highlighting the potential and the necessity of exploring the optimal implementation of cascading routers for specific applications.

### 5.4. RAG Results

Building on the results above, we simultaneously compared various router types, including predictive and cascading routers, on the RAG dataset. We used the same setting for KNN and MLP routers while selecting an error rate 0.2 for cascading routers. We randomly partitioned the RAG dataset into two splits: 70% for training predictive routers and 30% for evaluating all routers. Figure 6 demonstrates that all routers significantly improve compared to the Zero Router. Further analysis shows that the routers can identify time-sensitive features (like "2024") in user queries and route to online models for time-sensitive queries and GPT-4/GPT-3.5 for time-insensitive queries. Our findings highlight the potential of model routing to enhance LLM applications within the "Compound AI Systems" (Zaharia et al., 2024) scenario.

## 6. Limitations and Future Work

ROUTERBENCH currently only focuses on performance and economic cost. It is meaningful to include more evaluation criteria, such as latency, throughput, and others, to capture a more comprehensive understanding of router capabilities and limitations. There are also many LLMs and tasks that are not included in ROUTERBENCH due to the limitation of time, and future iterations of this benchmark would include datasets that cover more tasks to evaluate the ever-growing capability of LLMs effectively and also to add newer LLMs as they are being released.

Our current work only evaluates the efficacy of predictive and cascading routers, yet considerable scope exists for investigating further router designs, as highlighted in Section 5.3. Delving into more advanced router designs is crucial for enhancing routing efficiency. Notably, our evaluation within the RAG context was limited to models possessing inherent retrieval capabilities. Addressing the

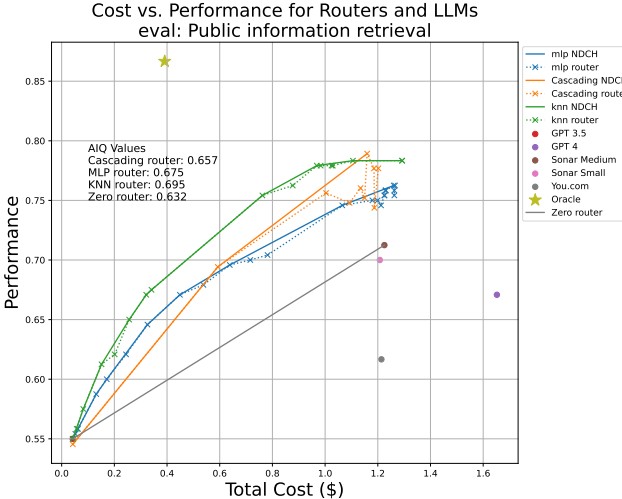

*Figure 6.* Total Cost vs Performance for five models and four routers on the RAG dataset. The AIQ values are also calculated for all four routers. NDCH represents the non-decreasing convex hull.

challenge of implementing two-stage routing, which encompasses retrievers and LLMs, remains critical. This approach could significantly refine router evaluations on standard RAG tasks, including HotpotQA (Yang et al., 2018) and NaturalQuestions (Kwiatkowski et al., 2019), by ensuring more precise assessments.

Furthermore, although the seven datasets in ROUTERBENCHoffer broad coverage across various tasks, incorporating domain-specific tasks that require long-tail skills, like translation of low-resource languages, could reveal additional intriguing aspects of LLM routing. This enhancement would align the benchmark more closely with real-world application scenarios. Efforts to integrate such tasks in future versions are planned.

## 7. Conclusion

We present ROUTERBENCH, a benchmark specifically designed to evaluate routers for multi-LLM systems. By addressing the critical need for standardized evaluation in this domain, our benchmark provides a comprehensive dataset and a theoretical framework designed for the nuanced analysis of router cost-efficiency and performance. The insights from our study shed light on the effectiveness of various routing strategies and revealed promising early results in some tasks. This work establishes a robust and scalable benchmark for router evaluation and aims to facilitate future progress in the efficient and cost-effective deployment of Large Language Models.

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

# A. Additional Dataset Details

## A.1. Model Details & Cost Estimation

For all proprietary models, we calculate the cost of input and output results based on their API pricing. For open-source models, we utilize Together AI [4] to obtain results and reference costs. For the RAG experiment, we refer to the API pricing of You.com [5] and Perplexity [6] for cost estimation.

## A.2. Dataset Details

**MMLU** (Hendrycks et al., 2021): A benchmark that measures the knowledge acquired by models during pretraining and evaluates models in zero-shot and few-shot settings across 57 tasks, testing both knowledge and reasoning on different fields of human knowledge.

**Hellaswag** (Zellers et al., 2019): This dataset challenges models to pick the best ending choice for a given sentence. It uses Adversarial Filtering(AF) to create a Goldilocks zone of complexity, wherein generations are largely nonsensical to humans but always make models struggle.

**GSM8K** (Cobbe et al., 2021): A dataset of diverse grade school math word problems, testing a model's ability to perform multi-step mathematical reasoning.

**ARC Challenge**(Clark et al., 2018) A rigorous question answering dataset, ARC-Challenge includes complex, different grade-school level questions that require reasoning beyond simple retrieval, testing the true comprehension capabilities of models. Arc Challenge dataset contains those that both a retrieval and a co-occurrence method fail to answer correctly)

**Winogrande** (Sakaguchi et al., 2021): A large-scale and increased harness dataset inspired by the original Winograd Schema Challenge(WSC) (Levesque et al., 2012) tests models on their ability to resolve pronoun ambiguity and their ability to understand the context with commonsense knowledge.

**MBPP** (Austin et al., 2021): The benchmark is designed to be solvable by entry-level programmers, covering programming fundamentals, standard library functionality, etc. Each problem comprises a task description, code solution, and 3 automated test cases.

**MT-Bench** (Zheng et al., 2023b): This dataset contains 3.3K expert-level pairwise human preferences for model responses generated by 6 models in response to 80 MT-bench questions, multi-run QA. The 6 models are GPT-4, GPT-3.5[7], Claude-v1, Vicuna-13B (Zheng et al., 2023a), Alpaca-13B (Taori et al., 2023), and LLaMA-13B (Touvron et al., 2023). The annotators are mostly graduate students with expertise in the topic areas of each of the questions. In this work, we only used the 80 questions to generate model responses for ROUTERBENCH.

## A.3. More Details on Dataset Construction

Each sample in the benchmark dataset will have the following attributes:

- $sample\_id$: contain the information about the name of the sub-task, the split of dataset, and the index of the data in that dataset. Example: **mmlu-astronomy.val.5**
- $model\_name$: the model used to perform inference for this sample. Example: **GPT-4**
- $eval\_name$: the source data from which this specific sample comes. Example: **hellaswag.dev.v0**
- $prompt$: prompt sentence. Example: **The following are multiple-choice questions...**
- $model\_response$: Model's output. Example: **The answer is A)**
- $performance$: the result compared to the true label. Example: **True/False**
- $cost$: for proprietary model, we use API cost to calculate; for open source model, we use Together AI[8] to call the model and use their cost as reference. Example: **0.00019**

---

[4]https://www.together.ai/pricing
[5]https://api.you.com/
[6]https://docs.perplexity.ai/docs/pricing
[7]https://openai.com/blog/chatgpt
[8]https://www.together.ai/

- *true_label*: the true label or gold response for this prompt. Example: **True/False**

## A.4. Evaluation Metrics

We will perform 5-shot inference on MMLU, HellaSwag, GSM8K, ARC Challenge, Winogrande, and 0-shot inference on MBPP, MT-Bench, and RAG.

For the datasets **MMLU**, **HellaSwag**, **GSM8K**, **ARC Challenge**, and **Winogrande**, we use the exact match method to compute the final results. In contrast, for **MBPP**, **MT-Bench**, and **RAG**, we use GPT-4 for answer evaluation. Results categorized as False/True are converted to a binary 0/1 format. In cases where the results are based on ratings, we normalize all outcomes to a [0, 1] scale.

## A.5. Individual Dataset Result

The ROUTERBENCH pilot study result has been shown in Figure 3. We provide the breakdown of each dataset in Figure 7. Additionally, we list the accuracies and costs for each individual model and the *Oracle* router in Table 1.

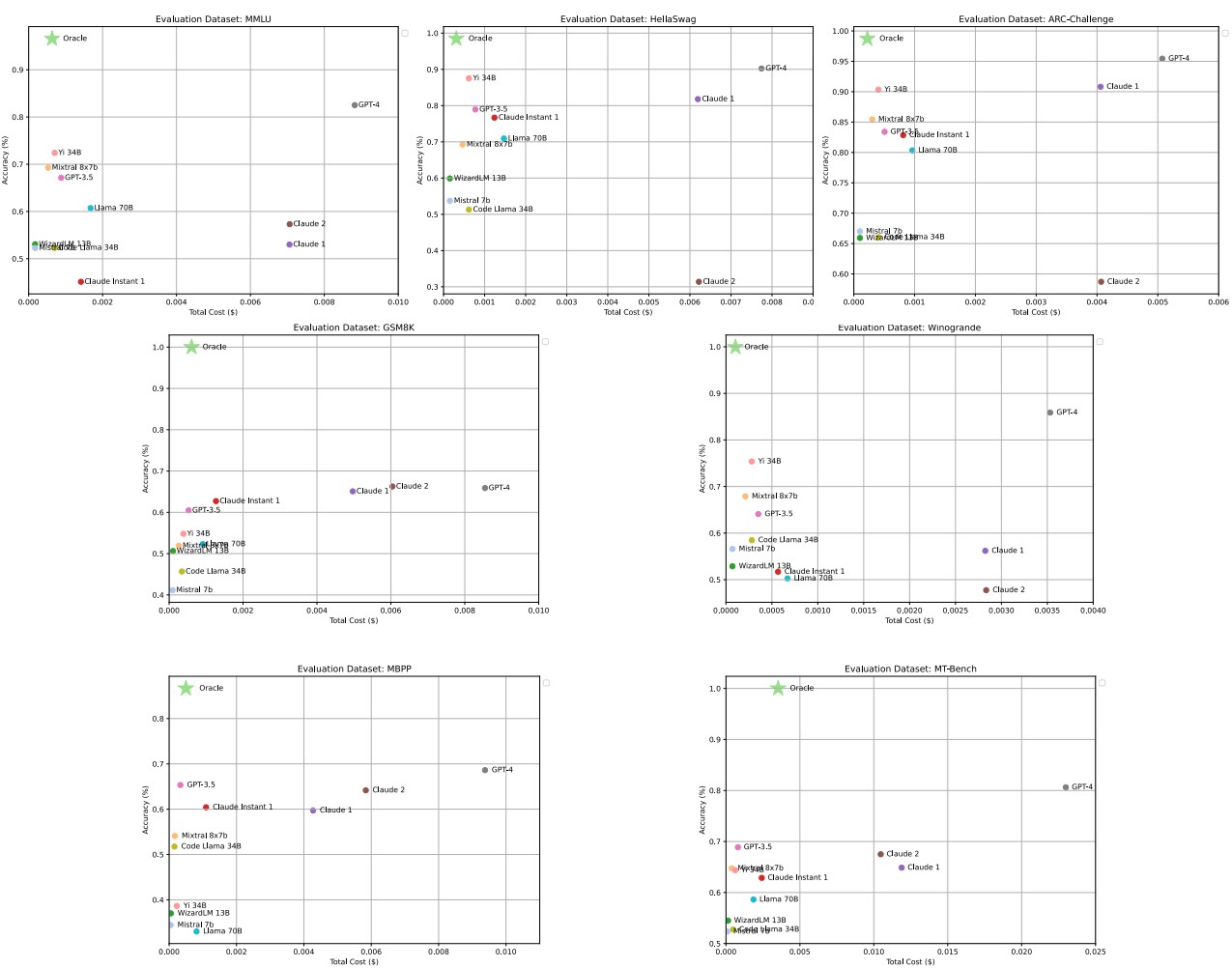

*Figure 7.* Accuracy vs Total cost of each LLM on each sub dataset in ROUTERBENCH.

Table 1. Individual models and the *Oracle* results on the seven datasets.

| Method | MMLU | | MT-Bench | | MBPP | | HellaSwag | | Winogrande | | GSM8k | | ARC | |
|---|---|---|---|---|---|---|---|---|---|---|---|---|---|---|
| | Perf↑ | Cost↓ | Perf↑ | Cost↓ | Perf↑ | Cost↓ | Perf↑ | Cost↓ | Perf↑ | Cost↓ | Perf↑ | Cost↓ | Perf↑ | Cost↓ |
| WizardLM 13B | 0.568 | 0.122 | 0.796 | 0.006 | 0.364 | 0.011 | 0.636 | 0.727 | 0.512 | 0.040 | 0.510 | 0.354 | 0.660 | 0.068 |
| Mistral 7B | 0.562 | 0.081 | 0.779 | 0.003 | 0.349 | 0.006 | 0.541 | 0.485 | 0.562 | 0.027 | 0.409 | 0.210 | 0.642 | 0.046 |
| Mixtral 8x7B | 0.733 | 0.245 | 0.921 | 0.012 | 0.573 | 0.023 | 0.707 | 1.455 | 0.677 | 0.081 | 0.515 | 0.594 | 0.844 | 0.137 |
| Code Llama 34B | 0.569 | 0.317 | 0.796 | 0.015 | 0.465 | 0.021 | 0.525 | 1.882 | 0.617 | 0.104 | 0.462 | 0.752 | 0.644 | 0.177 |
| Yi 34B | 0.743 | 0.326 | 0.938 | 0.018 | 0.333 | 0.031 | 0.931 | 1.938 | 0.748 | 0.107 | 0.552 | 0.867 | 0.882 | 0.182 |
| GPT-3.5 | 0.720 | 0.408 | 0.908 | 0.026 | 0.651 | 0.044 | 0.816 | 2.426 | 0.630 | 0.134 | 0.601 | 1.170 | 0.855 | 0.228 |
| Claude Instant V1 | 0.384 | 0.327 | 0.863 | 0.030 | 0.550 | 0.064 | 0.801 | 1.943 | 0.512 | 0.108 | 0.626 | 1.300 | 0.821 | 0.183 |
| Llama 70B | 0.647 | 0.367 | 0.854 | 0.022 | 0.302 | 0.039 | 0.736 | 2.183 | 0.504 | 0.121 | 0.529 | 0.870 | 0.794 | 0.205 |
| Claude V1 | 0.475 | 3.269 | 0.938 | 0.361 | 0.527 | 0.607 | 0.841 | 19.43 | 0.570 | 1.077 | 0.653 | 11.09 | 0.889 | 1.829 |
| Claude V2 | 0.619 | 3.270 | 0.854 | 0.277 | 0.605 | 0.770 | 0.421 | 19.50 | 0.446 | 1.081 | 0.664 | 13.49 | 0.546 | 1.833 |
| GPT-4 | 0.828 | 4.086 | 0.971 | 0.721 | 0.682 | 1.235 | 0.923 | 24.29 | 0.858 | 1.346 | 0.654 | 19.08 | 0.921 | 2.286 |
| *Oracle* | **0.957** | 0.297 | **0.996** | 0.052 | **0.899** | 0.041 | **0.994** | 0.860 | **1.0** | 0.042 | **0.748** | 1.282 | **0.977** | 0.091 |

## B. Extended Experimental Settings

We provide the hyperparameters of MLP and KNN routers in this section.

The KNN routers have two main hyperparameters that were tested in this paper. The number of neighbors, and the embedding model for the prompts. All KNN routers used cosine similarity as the distance metric, and used either 5, 10, or 40 neighbors. The embedding models were taken from the default SentenceTransformers library (Reimers & Gurevych, 2019), and are one of all-MiniLM-L12-v2, all-mpnet-base-v2, or all-distilroberta-v1. The best-performing hyperparameters for the KNN router were with 40 neighbors, and the all-MiniLM-L12-v2 embedding model.

In MLP routers, the models have either one or two hidden layers, with each layer having 100 neurons, and the ReLU activation function was applied. The learning rate was kept constant at 0.001, and the models took in embeddings from one of all-MiniLM-L12-v2, all-mpnet-base-v2, or all-distilroberta-v1. The best MLP router had two hidden layers of 100 neurons each, and used the all-MiniLM-L12-v2 embedding model.

## C. Issues with Overly-aligned Models

Some models exhibit reluctance in responding to certain inputs, often replying with statements like "I do not understand..." or "I am not sure about...". We have identified two primary reasons for models' refusal to respond:

**Insufficient Context Perception** Despite being provided with enough context, these models perceive the information as inadequate. Our hypothesis is that the models' capabilities might not be robust enough to generate answers or perform tasks effectively under these conditions. A potential remedy is to modify the prompting strategy to encourage output generation.

**Uncertainty Avoidance** Some models appear to be fine-tuned to function as 'safe' assistants, refraining from providing responses when they lack certainty. This cautious approach likely aims to prevent potential errors stemming from uncertain answers. Claude 2 exhibits this behavior most frequently.

LLMs have been known to have such kind of issues as documented in various previous studies (Zheng et al., 2024; Alzahrani et al., 2024). It is essential to apply methods that can make LLM outputs in a more controllable and structural way and automatically optimize their quality (Khattab et al., 2024; Singhvi et al., 2023) when routing, which warrants further exploration in future research.

## D. Full Cascading Routers Results

Here are we provide the rest cascading routers results on ARC-Challenge, MT-Bench, and HellaSwag.

## E. Training Data Distribution

We also conduct Out-domain experiments where we train on held-out tasks in ROUTERBENCH for each dataset and evaluate on MT-Bench, MBPP and GSM8K in Figure 9.

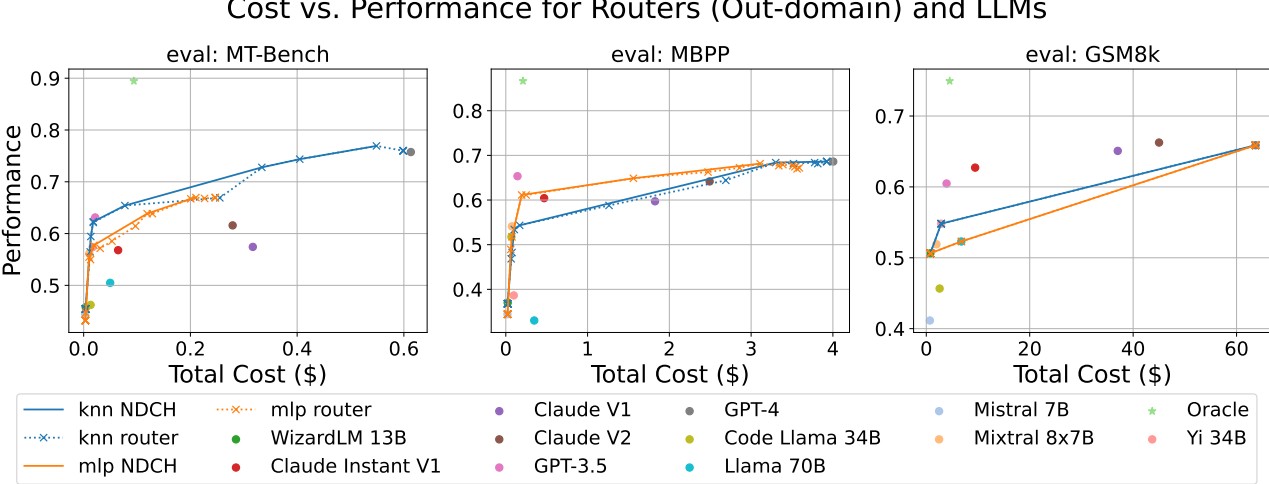

*Figure 8.* Total Cost vs Performance for eleven models and cascading routers on ARC-Challenge, MT-Bench, and HellaSwag. Different error rates are tested, and the AIQ value is computed for Zero Router and zero error rate cascading router. The solid lines represent non-decreasing convex hull and the dotted line represents points with increasing maximum cost parameter.

*Figure 9.* Total Cost vs Performance for eleven models and KNN, MLP routers on MT-Bench, MBPP, GSM8K. NDCH stands for non-decreasing convex hull

