# OpenReview forum: "RouterBench: A Benchmark for Multi-LLM Routing System"
_ICML.cc/2024/Workshop/Agentic_Markets — Agentic Markets @ ICML'24 Poster_

### Official Review · Reviewer_dum5 · 2024-06-10
**Review: RouterBench: A Benchmark for Multi-LLM Routing System**

**Rating:** 7
**Confidence:** 4

**Review:**

Summary:
The paper address the open problem of missing evaluation benchmarks for LLM routing systems by introducing a benchmark that evaluates factor like performance and inference cost across a diverse range of domains, models and tasks. The benchmark includes eight datasets and eleven distinct models. By providing 405,000 precomputed inference outcomes, it enables the development and evaluation of routing strategies without requiring repetitive real-time model inference.

Main contributions:
- The benchmark RouterBench to evaluate LLM routing systems
- Theoretical framework for the evaluation of such systems
- Comparative analysis w.r.t to cost and performance of models and routing systems, showing that costs can vary significantly (factor 2-5)

Pros:
- Addresses a significant research gap in the field
- Clear argumentation of design choices and strong theoretical framework is provided
- Extensive experimental section, offering new insights in the performance and cost of popular routing systems and models

Cons:
- Narrow scope w.r.t. metrics and factors that are evaluated, namely only the cost and performance (which is addressed in the limitation section)

Evaluation:
The paper is well written and addresses an important open problem. The presented benchmark is an original and meaningful contribution to the field, even though more work on the metrics and factors included in the benchmark, and covering a wider range of tasks would improve the benchmark significantly.

---

### Official Review · Reviewer_u6NV · 2024-06-15
**Novel and high-quality benchmark for an important problem**

**Rating:** 9
**Confidence:** 4

**Review:**

The work introduces a new benchmark for routers, something that didn't exist before, and evaluates it in-depth on 800 RAG queries and multiple contemporary benchmarks for various domains. 405,467 samples in RouterBench.

Strengths:
- Comprehensive dataset covering many domains
- Novel benchmark for an important problem in modern LM sampling
- Proper use of baseline routers and creation of new ones
- Great visualizations that explain relatively complex metric

Weaknesses:
- Would be great to see the distribution in RouterBench of various sources to understand if there's imbalance in the benchmark
- I wasn't sure what the quality metric exactly was but I assume it is something like performance compared to baseline given its range from 0 to 1 (probably an error in my reading)

Comments:
- No obvious grammatical errors